# Sarcopenia in Inflammatory Bowel Disease: A Narrative Overview

**DOI:** 10.3390/nu13020656

**Published:** 2021-02-17

**Authors:** Amritpal Dhaliwal, Jonathan I. Quinlan, Kellie Overthrow, Carolyn Greig, Janet M. Lord, Matthew J. Armstrong, Sheldon C. Cooper

**Affiliations:** 1Institute of Inflammation and Ageing, University of Birmingham, Birmingham B15 2WB, UK; a-dhaliwal@live.co.uk (A.D.); j.quinlan@bham.ac.uk (J.I.Q.); j.m.lord@bham.ac.uk (J.M.L.); 2Liver Unit, Queen Elizabeth Hospital, University Hospitals Birmingham NHS Foundation Trust, Birmingham B15 2TH, UK; mattyarm2010@googlemail.com; 3NIHR Biomedical Research Centre (BRC), University Hospitals Birmingham NHS Foundation Trust and University of Birmingham, Birmingham B15 2TH, UK; C.A.Greig@bham.ac.uk; 4Centre of Liver and Gastrointestinal Research, University Hospitals Birmingham NHS Foundation Trust and University of Birmingham, Birmingham B15 2TH, UK; 5School of Sport, Exercise and Rehabilitation Sciences, University of Birmingham, Birmingham B15 2TT, UK; 6Department of Gastroenterology, University Hospitals Birmingham NHS Foundation Trust, Birmingham B15 2TH, UK; Kellie.Overthrow@uhb.nhs.uk; 7MRC-Versus Arthritis Centre for Musculoskeletal Ageing Research, University of Birmingham, Birmingham B15 2WB, UK

**Keywords:** sarcopenia, inflammatory bowel disease, muscle mass, exercise, nutrition

## Abstract

Malnutrition is a common condition encountered in patients with inflammatory bowel disease (IBD) and is often associated with sarcopenia (the reduction of muscle mass and strength) which is an ever-growing consideration in chronic diseases. Recent data suggest the prevalence of sarcopenia is 52% and 37% in Crohn’s disease and ulcerative colitis, respectively, however it is challenging to fully appreciate the prevalence of sarcopenia in IBD. Sarcopenia is an important consideration in the management of IBD, including the impact on quality of life, prognostication, and treatment such as surgical interventions, biologics and immunomodulators. There is evolving research in many chronic inflammatory states, such as chronic liver disease and rheumatoid arthritis, whereby interventions have begun to be developed to counteract sarcopenia. The purpose of this review is to evaluate the current literature regarding the impact of sarcopenia in the management of IBD, from mechanistic drivers through to assessment and management.

## 1. Introduction

Malnutrition, defined as ‘a state resulting from the lack of intake or uptake of nutrition that leads to altered body composition and body cell mass leading to diminished physical and mental function and impaired clinical outcome from disease’ [1], is frequently observed in patients with inflammatory bowel disease (IBD). Unfortunately, malnutrition is known to be a key driver for muscle loss and the consequential loss of function [1], a condition known as sarcopenia. Sarcopenia was first described by Rosenburg in 1988 and was defined as the age-related loss of muscle mass [2], although it is clear that sarcopenia can also occur as a result of chronic illness, inactivity and inflammation [3]. More recent definitions of sarcopenia encompass the reduction of muscle strength alongside the reduction of muscle mass and in which poor physical performance is indicative of severe sarcopenia [4].

IBD includes both ulcerative colitis (UC) and Crohn’s disease (CD). UC is an idiopathic chronic inflammatory disorder which targets the colonic mucosa, usually in a continuous pattern, extending to varying degrees from the rectum, proximally to the remaining colon. Whereas CD is a transmural, progressive inflammatory disease and can affect any segment of the gastrointestinal (GI) tract. As such, complications including strictures, fistulae and abscesses occur more frequent in CD [5]. IBD has an unpredictable clinical course with undulating periods of acute and chronic exacerbations with remissions [5,6]. Currently there are no curative treatments for CD, whereas UC can be cured by a pan proctocolectomy.

Significantly, a greater degree of malnutrition can be seen in patients with CD compared to UC (75% and 62%, respectively) [7]. This may be due to the impact of CD on the main site for nutrient absorption (i.e., enteropathy), or the persistent nature of CD [8], with the sequelae from fistulation, short bowel syndrome following resections, or partial obstructive symptoms secondary to stricturing disease. Therapies for targeting inflammation are often the focus of current research, with less focus on the prevalence and impact of sarcopenia in IBD; despite a recent meta-analysis showing that 52% of patients with CD and 37% of patients with UC have sarcopenia [9]. These results are significant as sarcopenia is known to have a clear impact on quality of life, length of hospital stay, surgical outcomes, and mortality [10,11,12].

Sarcopenia is, thus, an important consideration in the management of IBD, in both prognostication and treatment. Whilst there is evolving research about interventions to overcome sarcopenia in many chronic inflammatory states, such as chronic liver disease and rheumatoid arthritis [13,14,15,16], the same cannot yet be said for IBD. The purpose of this narrative review is to evaluate the current literature, regarding the assessment of sarcopenia, mechanistic drivers behind sarcopenia in IBD and, finally, the management of sarcopenia in an IBD population.

## 2. Methodology

Doctors A. Dhaliwal and J. Quinlan, and Mrs K. Overthrow (AD, JQ and KO) searched for the relevant research conducted in patients with inflammatory bowel disease. The online databases MEDLINE (Pubmed) and Cochrane were searched for eligible studies. Key search terms, including medical subject headings (MeSH) terms, used were: sarcopenia, inflammatory bowel disease, crohn’s disease, ulcerative colitis, muscle mass, exercise, physical activity, dietary supplements, exercise. All studies underwent preliminary screening by reviewing their titles and abstract. Full text articles were then reviewed prior to inclusion. A final review of the references was undertaken to ensure appropriate studies were sourced and included [17,18].

## 3. Assessment of Sarcopenia

### 3.1. Muscle Mass

The original definition of sarcopenia was solely focused upon muscle mass loss [2]. As such, the majority of research quantifying sarcopenia in IBD has assessed only muscle mass [19,20,21,22]. Typically, two imaging techniques have been employed for the assessment of muscle mass: computed tomography (CT) and dual energy X-ray absorptiometry (DXA). It is important to note that CT allows a direct estimate of muscle mass, whereas DXA only provides indirect estimates such as lean mass. Nonetheless, cross sectional abdominal CT imaging is often part of routine assessment of IBD and thus no additional scans are required [19,20,21,23,24]. Once obtained, the total muscle cross-sectional area (CSA) is obtained at the L3-vetebra level, although in some circumstances muscle density has also been estimated via the Hounsfield Unit average calculation (HUAC) [21]. Nonetheless, values of the 3rd lumbar vertebrae (L3) muscle CSA are often converted into a skeletal muscle index (SMI). It is from these data that Zhang et al. and Lee et al. generated L3 SMI cut offs for sarcopenia of 52.4 cm^2^/m^2^ and 38.54 cm^2^/m^2^ [20] and 49.4 cm^2^/m^2^ and 31.4 cm^2^/m^2^ [19], for males and females, respectively. It should be noted that both studies completed their work in an Asian population; thus, these cut offs may differ in western populations. The other, less-utilised technique is DXA and the generation of appendicular skeletal muscle index (ASMI) [22,25]. Similar to Lee and Zhang, Schneider and colleagues also generated cut off values for sarcopenia (as ASMI as opposed to SMI) of 7.26 kg/m^2^ and 5.45 kg/m^2^ for males and females respectively. However, regardless of technique, there seems to be a general agreement that sarcopenia (when assessed via muscle mass) is prevalent in ~40–60% of IBD patients [19,20,24,25]. Further, as expected when IBD patients are categorised into either UC or CD; the greater incidence of sarcopenia occurs in CD [8,21].

### 3.2. Physical Function

In contrast to the numerous studies which investigate sarcopenia with respect to muscle mass; the same cannot be said for muscle function. Unlike age-related sarcopenia [4], no specific cut offs for functional assessments such as handgrip, gait speed or timed up and go (also known as chair stands) have been formulated in an IBD population. As such, it is likely that there is a piece of the puzzle missing in counteracting sarcopenia in IBD patients. One easy and reproducible test which can readily be incorporated into the routine assessment is handgrip dynamometry. Handgrip strength has long been known to strongly correlate with lean mass and sarcopenia [26]. Indeed, a thorough assessment of sarcopenia, inclusive of function, is essential; particularly as reduced function may have a negative impact upon post-operative outcomes [9].

### 3.3. Nutrition

Skeletal muscle is a highly metabolic tissue which is maintained through acute anabolic signals such as exercise and nutrition [27]. As such, nutritional assessment plays a fundamental role in management of IBD as patients incur a high risk of nutritional depletion and malnutrition. The aetiology of malnutrition in IBD is multifactorial, and is evidenced by protein-energy malnutrition, alterations in body composition and (micro)nutritional deficiencies. In addition, elevated nutritional requirements exist due to the metabolic cost of inflammation and mucosal tissue repair, and pharmacological therapy [7]. Consequently, malnutrition is routinely screened in IBD patients using a nutritional screening tools such as “MUST” (Malnutrition University Screening Tool), NRS (Nutritional Risk Screening)—2002, NRI (Nutritional Risk Index), MIRT (Malnutrition Inflammation Risk Tool), and the SaskIBD-NRT (Saskatchewan Inflammatory Bowel Disease Nutrition Risk Tool); evidence validating these in the use of IBD requires further research [28]. These traditional screening tools may place too much emphasis on body mass index (BMI) and weight loss, which are believed to be unreliable indicators of malnutrition in IBD patients [29]. Indeed, a recent review by Adams et al. [24] found that whilst almost half of their IBD study cohort were sarcopenic (49%), the large proportion of these patients were of normal weight (41.4%), overweight or obese (19.5%).

Sarcopenic obesity, which is characterised by a relative reduction in muscle mass due to increased fat mass [30], is increasing in parallel with the obesity epidemic. Sarcopenic obesity can be linked to a fast functional decline in patient status, with a higher risk of disability, mobility and mortality [7]. Furthermore, sarcopenic obesity has been recently identified as a negative predictor of survival in cancer patients, liver cirrhosis, and cardio-metabolic disease [24]. Significantly, obesity may mask signs of poor nutritional status in IBD, such as loss of lean mass and bone mineral loss [31]. The prevalence of obesity in IBD is estimated to be between 15% and 40% [32]; as such obesity is an important consideration for health care practitioners when identifying malnutrition in IBD.

## 4. Mechanisms Driving Sarcopenia in IBD

Muscle mass is tightly regulated by the balance of muscle protein synthesis (MPS) and muscle protein breakdown (MPB). Muscle mass is maintained when MPS = MPB, atrophy occurs when MPB > MPS and hypertrophy when MPS > MPB. In the context of sarcopenia, it is believed that an imbalance exists whereby protein metabolism favours breakdown [33]. There are many possible drivers behind this imbalance and the consequential development of sarcopenia. This review will attempt to elucidate some of the prominent mechanisms of sarcopenia within IBD, including inflammation, vitamin D deficiency, adiposity and malabsorption before finally discussing the muscle-gut axis. Other factors such as reduced physical activity and specific drug treatment should also be considered, however these are beyond the scope of this review.

### 4.1. Chronic Inflammation

Chronic inflammation is believed to be a key driver in sarcopenia in IBD. IBD is accompanied by a systemic increase in circulatory proinflammatory cytokines such as interferon gamma (IFNγ), interleukins (IL)-1, 6, 12, 18 and in particular tumour necrosis factor alpha (TNFα) [34]. Inflammatory cytokines have a direct catabolic effect on protein metabolism, reducing MPS and anabolic drive [35,36]. Indeed, TNFα triggers MPB via the inhibition of the anabolic mammalian target of rapamycin complex 1 (mTORC1) pathway, whilst also stimulating atrogenes muscle atrophy box (MAFBx) and muscle ring finger-1 (MURF-1) which in turn increase MPB [37]. TNFα also increases reactive oxidative stress (ROS) which activates the nuclear factor kappa-light-chain-enhancer of activated B cells (NF-κB) pathway, which in turn enhances MPB, reduces myogenesis and triggers further downstream inflammation [38,39]. NF-κB can be inhibited by peroxisome proliferator-activated receptor gamma (PPAR-γ). However, in IBD PPAR-γ expression is believed to be reduced; thus a reduced ability to inhibit NF-κB exists [40], further adding to MPB. TNFα is also able to activate the enzyme 11 beta-hydroxysteroid dehydrogenase type 1 (11-βHSD1) in muscle which converts inactive cortisone to catabolic cortisol, contributing to MPB [41].

In addition to activating NF-κB and the ubiquitin proteasome system [7], proinflammatory cytokines are also known to reduce levels of plasma and muscle Insulin like growth factor 1 (IGF-1) [42]. The decrease in IGF-1 results in a reduced activation of the protein kinase B aka AKT and phosphoinositide 3-kinase (PI3K) pathways and in turn reduces stimulation of mTORC1 and consequently MPS. NF-KB and TNF-α are able to induce upregulation of the MPS inhibitor myostatin, which inhibits mTORC1 and activates MURF-1 and MAFBx [43,44]. All of the above favour MPB and reduce MPS, leading to muscle loss and sarcopenia [45] (Figure 1).

### 4.2. Vitamin D Deficiency

Vitamin D is a fat-soluble vitamin which is primarily produced in the skin when exposed to ultraviolet light but is also present in small quantities in certain foods, e.g., oily fish and eggs. The role of Vitamin D within skeletal muscle and sarcopenia has gained increased attention [46]. Vitamin D is thought to have many important roles within skeletal muscle including maintaining contractile excitability via intracellular calcium, muscle stem cell proliferation and differentiation and the consequential maintenance of muscle function [46]. Unsurprisingly, a deficiency or even an insufficiency of vitamin D is correlated with the risk of diseases including sarcopenia, cardiovascular disease, obesity and cancer [47]. According to the UK Scientific Advisory Committee on Nutrition (SACN), vitamin D deficiency is classified as serum levels of 25-hydroxyvitamin D (25-OH-D) below <25 nmol/L. However, others suggest that deficiency is serum levels of 25-OH-D below 25–50 nmol/L, with sufficiency >75 nmol/L and hence insufficiency in the region of 51–74 nmol/L [48,49,50,51]. Nonetheless, vitamin D deficiency is frequently observed in patients in IBD [52] and is suggested to be between 30 and 47% [52]. Significantly, vitamin D deficiency is associated with sarcopenia in older people [46]. Whilst the specific mechanisms whereby vitamin D deficiency contributes to muscle loss are unclear; it is likely that the vitamin D receptor (VDR) plays a role via changes in anabolic signaling, muscle protein synthesis and translational efficacy [53,54]. Aside from the direct actions on skeletal muscle, the VDR is also known to impact mitochondrial function, whereby depletion of the VDR results in reduced oxidative phosphorylation output [55]. In addition, mitochondrial dysfunction can result in the increased production of reactive oxygen species, which are known to have a negative impact on skeletal muscle and hence contribute to sarcopenia [56]. Vitamin D also has a role in preserving the integrity of mucosal barriers in inflammatory disease and its homeostasis by modulating immune and inflammatory responses, in particular via negatively impacting macrophage function [52,57]. Given the damage to mucosa in IBD, Vitamin D deficiency may further increase susceptibility to mucosal damage, increase severity of IBD and potentially enhance sarcopenia [57].

### 4.3. Adiposity

Adipose tissue and specifically the hormones secreted by adipocytes, adipokines, are known to be an integral part in the homeostasis of energy metabolism; however, it is also involved in immune system regulation and muscle protein homeostasis [58]. Dysregulation of adipose tissue is observed in many chronic inflammatory conditions and is “characterized” by low grade inflammation and increased amount of both central and ectopic adipose deposition [7]. In IBD, more so in CD than UC, an increase in visceral adiposity and in particular mesenteric fat, has been observed in cross-sectional imaging [59]. The term “creeping fat” has been used to describe mesenteric adipose that encompasses greater than half of the circumference of the inflamed intestinal region [59]. This allows the use of radiological evidence to differentiate from active inflammation and fibrosis. Mesenteric fat may not necessarily correlate with BMI as it can occur in increased inflammatory states in the absence of a raised BMI; therefore, it can frequently be overlooked with a normal or low BMI. Standard cross-sectional imaging can provide a rudimentary assessment of presumed BMI. Additionally mesenteric adipocytes produce inflammatory cytokines (i.e., IL-6, TNFα) and release free fatty acids that further contribute to systemic inflammation with the detrimental effect on muscle mass described above, and also interfere with intestinal homeostasis [60]. Research suggests that the mesenteric adiposity has some correlation with the overexpression of these proinflammatory cytokines, and consequently CD severity and muscle wasting [7,61].

### 4.4. Malabsorption

Malabsorption is often a direct consequence of mucosal alterations, such as impaired epithelial transport and loss of epithelial integrity. Active intestine inflammation releases inflammatory mediators (e.g., TNFα) which increase the permeability of the intestinal membrane and cause both local and systemic inflammatory effects [62,63]. Inflammation increases GI transit, thereby reducing the contact time between nutrients and the intestinal surface; this worsens malabsorption and reduces amino acid absorption [62]. This in turn will enhance sarcopenia, as amino acids are a major anabolic signal for muscle.

Malabsorption can be further exacerbated by a condition known as small intestinal bacterial overgrowth (SIBO); whereby an increase to the normal bacterial population occurs in the GI tract. SIBO is related to chronic intestinal inflammation, stricturing, dysmotility and partial chronic obstruction, and is known to alter intestinal permeability through similar mechanisms [62,63]. Further malabsorption may occur due to bowel resections, whereby the mucosal surface area available for absorption of nutrients is consequently reduced, leading potentially to short bowel syndrome and, in severe cases, intestinal failure. Other causes of malabsorption such as vomiting, diarrhoea, loss of appetite, anorexia and side effects of drugs, e.g., pancreatitis, are also contributors [63,64].

### 4.5. Muscle Microbiome Axis

The muscle–gut axis plays an integral role in muscle homeostasis. Healthy gut microbiota regulates immune and metabolic homeostasis, and gene expression via the production of short chain fatty acids (SCFA) and antioxidants which in turn influence resident mucosal immune cells to produce proinflammatory cytokines [65,66]. Additionally it has been suggested that the gut microbiome has a direct modulatory effect on amino acid bioavailability [67]. Some gut bacteria are derived from oral intake which plays an integral role in the mass and diversity of gut microbiota [65,66]. Environmental and social factors such as geographical location and regional food preferences also play a role as they can affect dietary composition thus altering gut microbiota [65,68]. The gut microbiota transduces signals for nutrients to produce mediators for muscle homeostasis [69]; for example SCFA act on skeletal muscle mitochondria and can influence MPS [70]. However, low SCFA production has been associated with increased subclinical chronic inflammation and an enforced degree of anabolic resistance [70,71]. There are several bacterial taxa that have been identified for their potential role in skeletal muscle functions [72]. For example, *Bifidobacteria lactobacilli* is involved in facilitating protein breakdown to amino acids within the gut, producing SCFA for energy production. stimulation of IGF-1/mTORC1 pathway and promoting the expression of genes involved in MPS [73,74,75]. *Escherichia coli* and *Klebsiella* have roles in the stimulation of anabolism and cell proliferation by further stimulating the IGF-1/mTORC1 pathway. These processes stimulate MPS and, hence, may counteract the development of sarcopenia. It is important to note that these bacteria need nutrient substrates such as folate and vitamin B12 for maintenance of their function; therefore, nutrient deficiency will negatively impact the function of the gut microbiota [69]. Given the potential involvement in muscle protein homeostasis, further research exploring the influence of gut microbiome on sarcopenia is required.

## 5. The Management of Sarcopenia

### 5.1. Nutrition

There is an ever-growing body of evidence that supports the use of nutritional intervention to help support muscle mass and consequently muscle strength and function in those with age-related sarcopenia [76]. Similarly, the optimisation of nutritional status may offer a potent tool in counteracting sarcopenia in IBD. However, there is currently no general consensus on the role of specific diets and supplementation for the prevention and treatment of sarcopenia in IBD.

One key macronutrient for counteracting sarcopenia is dietary protein. Amino acids are an anabolic stimulus which can trigger acute spikes in MPS which offset MPB, resulting in a net balance of protein turnover. It is known that lowered protein intakes exist in older adults [77] and it is believed this likely contributes to sarcopenia due to an imbalance of MPB and MPS (i.e., MPB > MPS). Similarly, in IBD patients, higher protein doses of 1.2–1.5 g/kg/d are often recommended during active disease states [10] in an attempt to prevent muscle loss. These higher levels of protein intake may be achieved with tailored dietary advice to include oral nutritional supplements (which will be discussed later) and food fortification.

Unsurprisingly, patients with malnutrition or sarcopenia may benefit from pre-operative nutritional optimisation. Indeed, one study reported fewer major post-operative complications in patients with a high Nutritional Risk Score (i.e., >3) who had preoperative nutritional support compared to those that did not (6.5% vs. 28.6% respectively) [20]. Due to the reduced risk of major post-operative complications seen with perioperative nutritional management, dietetic and physiotherapy assessment in all pre-operative IBD patients could be beneficial [9].

### 5.2. Supplementation

There is a considerable wealth of literature which investigates supplemental interventions for the prevention of muscle loss, strength and function in older adults [78,79]; however, the same does not exist within those suffering with IBD. This is particularly surprising as reduced intake due to malabsorption may place greater importance on supplementation. Nonetheless evidence from age-related sarcopenia may provide potential avenues for counteracting sarcopenia in IBD, including oral nutritional supplementation.

Whilst there is a large body of supporting evidence for the use of enteral and parental nutrition in patients with inflammatory bowel disease, the data on its effect on sarcopenia in patients with IBD are lacking at present.

For the purposes of this review, we discuss three main routes of supplementation below: supplemental protein, vitamin D3 and omega 3 polyunsaturated fatty acids (*n*-3 PUFAs).

### 5.3. Protein

Protein or specifically amino acids are essential for the maintenance of muscle tissue and may have increased importance with ageing [80] and chronic disease [81]. Indeed, protein intake and/or absorption is likely reduced in IBD and, thus, oral supplements in the form of protein supplementation may help to mitigate this deficit. Further to reduced dietary protein intake, it is possible that a phenomenon known as ‘anabolic resistance’ may occur. Anabolic resistance is seen in age-related sarcopenia such that a reduced acute response (in terms of reduced molecular activation or dampened MPS) to anabolic stimuli such as exercise or protein ingestion occurs. Over a chronic period, these dampened responses are believed to contribute to muscle loss and sarcopenia [82]. The mechanistic drivers of anabolic resistance are still poorly understood and likely multifactorial [83]; however, it is believed that chronic low-grade inflammation is a likely contributor [84]. Indeed, one previous study demonstrated reduced levels of total and phosphorylated Akt, a key site in the MPS pathway, in patients with CD compared to controls [85]. Unfortunately, food intake was not controlled for in this study and so participants could have been in either the fasted or postprandial state thus causing conflicts in the data. Nonetheless, if some degree of anabolic resistance occurs in IBD then data in elderly individuals suggest that larger acute doses of protein may be able to overcome this [82]. However, to our best knowledge, no previous study has investigated the impact of protein supplementation on muscle health within an IBD population.

### 5.4. Vitamin D

Several studies have highlighted the benefits of vitamin D replacement in patients with IBD, however, there are limited studies identifying the effect of vitamin D supplementation in IBD on muscle mass or function. Vitamin D supplementation in adults improves muscle strength in pre-sarcopenic elderly patients [86], healthy 18- to 40-year-old [87] and post-menopausal cohorts [88]. Interestingly, vitamin D supplementation in children aged 9–15 years with IBD has shown an improvement in maximal muscle power measured by jumping mechanography at [89], but data in adults are lacking.

### 5.5. Omega 3 Polyunsaturated Fatty Acids

*n*-3 PUFAs are a family of lipids important for health and exist in several forms that include alpha-linolenic acid (ALA), docosahexaenoic acid (DHA), eicosapentaenoic acid (EPA) and docosapentaenoic acid (DPA). ALA is an essential short chain PUFA which must be obtained via diet, whereas EPA and DHA can be synthesised from PUFAs. PUFAs are believed to possess anti-inflammatory effects [90] due to their ability to increase the expression of peroxisome proliferator-activated receptor gamma (PPAR-γ) and, in turn, inhibit proinflammatory NF-κB. Significantly, a reduced expression of PPAR-γ has been reported in those suffering from CD [91] and the consequential increase in NF-κB is suggested as a cause of IBD [38,92]. Therefore, the premise exists for the use of PUFAs in treating sarcopenia, specifically within patients with IBD. Indeed, PUFAs are postulated to reduce MPB in many disuse and disease states, such as cancer cachexia, muscle-disuse and pre/post-surgery [93] However, a systematic review and meta-analysis suggested that while omega 3 PUFAs may increase bone mineral density in elderly individuals, it has no clear effect on skeletal muscle or functional outcomes [94]. Nonetheless, this review did include a large range of clinical conditions as well as healthy adults; thus a true effect in IBD may have been masked. Therefore, further high-quality randomized controlled trials (RCTs) with longer duration of supplementation are needed to establish their effect and specifically in an IBD population.

### 5.6. Physical Activity

Physical activity, both aerobic and resistance exercise training (RET), are well established methods to improve cardiovascular health [95], muscle mass and muscle strength [96]. Therefore, it is no surprise that physical activity has drawn significant attention in many chronic inflammatory conditions [97,98,99,100]. Physical activity is becoming increasingly recognised as possessing anti-inflammatory effects, primarily through the emerging role of anti-inflammatory myokines, such as IL-6 which is released into the circulation in high amounts during exercise [101,102]. However, in comparison to healthy individuals, physical activity participation presents concerns for those suffering from IBD [103]. Although many studies have demonstrated the safety and efficacy of physical activity within the IBD patient group [104,105,106], these studies generally involved individuals who suffered from a mild disease state or were in remission. Nonetheless, the large majority of physical activity intervention studies have focused on low- or moderate-intensity aerobic exercise [104,106,107,108] with the addition of motivational intervention and stress management [107,109]. Unsurprisingly, exercise was found to induce positive changes in the Health related quality of life (HRQOL) and Inflammatory bowel diease questionnaire (IBDQ) scores (questionnaires for quality of life and disease severity, respectively) [104,109] and in particular several authors noted an improved mental state [107,110]. In regard to changes in immune function, one study demonstrated that neutrophils, lymphocytes, monocytes, IL-6 and IL-17 all increased in response to a single 30-minute bout of moderate cycling [105]. However, following 10 weeks of moderate exercise no long-lasting changes in immune function were observed [107]. Therefore, it is possible that the anti-inflammatory effects of exercise in IBD may be due to repeated acute bouts. For a more detailed review see reference [100]. Nonetheless, what is striking is the lack of measurement for key variables such as muscle mass and muscle strength in these studies, despite the definition of sarcopenia incorporating both of these elements [4]. Consistent with this, is a paucity of studies which investigate the impact and efficacy of RET in IBD. RET in particular is well known to induce positive changes to both muscle mass and strength, even in frail individuals [111]. One early study did investigate the effects of 12 months of low impact exercises via resistance bands within an IBD population [112]. The authors focused in particular upon the impact on bone mineral density, which increased at the trochanter following exercise. However, similar to those studies utilising aerobic exercise, there was no measure of muscle mass or strength. Therefore, it is clear that there is a knowledge gap within the literature such that the safety and efficacy of a RET intervention within an IBD population should be investigated as a means to counteract sarcopenia.

Furthermore, in recent years ‘prehabilitation’ has resulted in reduced lengths of hospital stay and 5-year disease-free rates in patients undergoing colorectal surgery, albeit for those with underlying malignancy (Trepanier 2019). Prehabilitation is a multi-modal (nutrition, exercise, psychological) intervention to enhance functional capacity in anticipation of a forthcoming physical stressor (i.e., surgery). Studies have largely focused on middle-aged patients with colorectal cancer, but the concept has the potential to have a significant impact on those patients with refractory IBD awaiting (semi)elective surgery. The type, duration and setting (supervised vs. remote learning) of prehabilitation in patients with IBD warrants further research.

### 5.7. Pharmacological Treatment

There have been several advancements in medical treatment for IBD with biologic agents such as anti-TNFα agents and newer anti-interleukin and anti-integrin agents, both of which demonstrate clear efficacy in the treatment of IBD and reducing inflammation [113]. These medications are used in conjunction with or following the failure of first line treatments such as corticosteroids and thiopurine immunomodulators (azathioprine and 6 Mercaptopurine) [11]. Unsurprisingly, the primary action of anti-TNFα agents is through the inhibition of TNFα to reduce inflammation resulting in the reduction of catabolic signalling, apoptosis and cytotoxicity [114]. Anti-integrin agents antagonise the action of integrins, thus reducing the movement of immune cells to the intestinal endothelium and suppressing the recruitment of inflammatory cells to damaged intestinal mucosa, whilst the anti-interleukins block the action of IL-12 and IL-23 [115]. These treatments combat inflammation and, in turn, may reduce the catabolic effects on skeletal muscle and, therefore, could potentially improve sarcopenia.

Unfortunately, there are limited studies that investigate the direct effect of biologics on sarcopenia in an IBD cohort. Subramaniam et al. [116] is the only prospective study measuring sarcopenia pre and post anti-TNFα treatment for patients with an acute flare of CD. They observed improvement in both quadriceps muscle volume and maximal isokinetic quadriceps strength from baseline after 25 weeks of infliximab treatment [116]. This was supported by Csontos et al. [117] who observed an improved BMI and muscle parameters in the form of fat free mass index, following a 12-week observational period in an outpatient cohort of both CD and UC commencing either Adalimumab or Infliximab.

Furthermore, there are other benefits of anti-TNFα agents in improving body composition. Several studies observed increases in body weight, lean mass and BMI [118,119,120,121], strongly suggesting an improvement in muscle mass with the treatment of active inflammation. However, it is important to note that some also noticed an increase in fat mass [122,123] which may suggest that anti-TNF treatment may also play a role in increasing adiposity.

One final point of consideration is the concomitant use of corticosteroids, however short-term, with anti-TNFα, anti-interleukin or anti- integrin agents; therefore, it is challenging to delineate the direct effects of each. The effect of corticosteroids on muscle homeostasis is shown to simultaneously improve inflammation [124,125] but also elicit greater rates of MPB [126]. Further studies are required to determine the impact of anti-TNFα, anti-interleukin and anti-integrin agents and other drugs used in the treatment of IBD, on sarcopenia in terms of muscle quantity, quality and performance.

### 5.8. Surgical Intervention

Many patients with IBD undergo surgical treatment. A range of surgeries are utilised to manage IBD, ranging from stricturoplasty to resections such as small bowel resections and pan-proctocolectomy resulting in permanent stoma formation; all will have an impact of muscle preservation [127]. Hence it is important to understand the role of surgical intervention in managing sarcopenia. A systematic review completed by Ryan et al. [9] reviewed existing data in IBD and sarcopenia with its correlation with surgical outcomes. It was found that IBD patients with sarcopenia had a higher probability of requiring surgery and significantly higher rates of major complications [9]. There are a number of studies that support these findings in terms of sarcopenia as a predictor of surgical intervention and reduced operation-free survival [20,24,128,129]. Indeed, there is an increase in post-surgical infections in those with sarcopenia, specifically surgical-site infections after restorative proctocolectomy for UC [130] and abscesses following intestinal surgery in CD [131]. However, in addition to muscle mass, an emerging consideration is muscle quality, more specifically the effect of myosteastosis. O’Brien et al. [132] observed that myosteatosis was significantly associated with increased hospital stay postoperatively (13 days vs. 9 days) and an independent association between myosteatosis and hospital readmission existed (odds ratio (OR) = 4.802).

Research has shown that, with treatments such as anti-TNF agents [118], sarcopenia and disease activity can be reversed. To the best of our knowledge, no prospective studies exist to identify the outcome of sarcopenia or muscle mass following surgical intervention in IBD. Zhang et al. [8] identified that in a retrospective study of patients with UC who underwent surgery (ileal pouch-anal anastomosis or colectomy with permanent ileostomy); the prevalence of sarcopenia and disease activity (via mayo scoring (a scoring system which evaluates UC, based on four parameters: stool frequency, rectal bleeding, endoscopic evaluation and clinical assessment) was reversed pre and post surgery. The authors demonstrated that sarcopenia was a positive predictor of high mayo score (OR, 8.49) and that sarcopenia and SMI in UC increases the probability of colectomy [8]. However, further research into the effect of surgical intervention on sarcopenia in IBD is required (Figure 2, Table 1).

## 6. Conclusions

Patients with IBD are at an increased risk of developing sarcopenia. It is clear that sarcopenia has a negative impact on length of hospital stay, surgical outcomes and clinical course. Indeed, sarcopenia can be a strong predictor of outcomes such as the need for surgical intervention and post-operative complications. However, a large degree of heterogeneity exists in the assessment of sarcopenia in IBD cohorts, with a predominance in focusing on muscle mass parameters only. There is a need to incorporate muscle strength and physical function measures into the identification of sarcopenia and its impact on physical performance in IBD cohorts. We surmise that further research is required to incorporate recognition of sarcopenia into the standard nutritional assessment and to utilise this to tailor individual management. Finally, further studies pertaining to treatment effects are required (medical, surgical, nutritional supplementation and exercise) to fully comprehend the role intervention plays in mediating sarcopenia in IBD.

## Figures and Tables

**Figure 1 nutrients-13-00656-f001:**
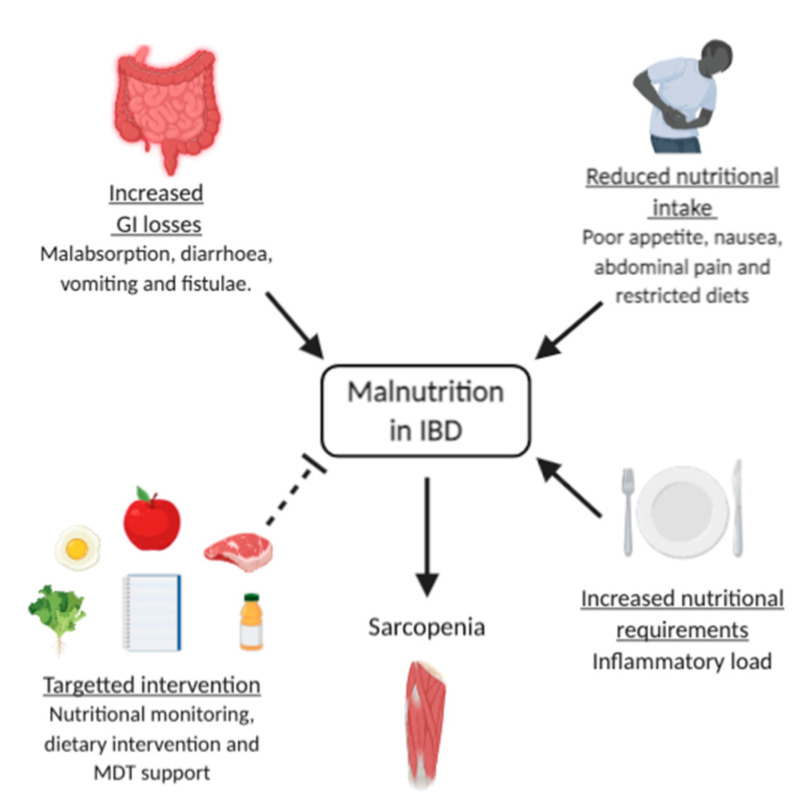
Multi-factorial causes of malnutrition and their contribution to sarcopenia. Solid arrows reflect negative contributors to malnutrition whereas dashed lined represents intervention and preventative strategies. Inflammatory bowel disease (IBD), multidisciplinary team (MDT).

**Figure 2 nutrients-13-00656-f002:**
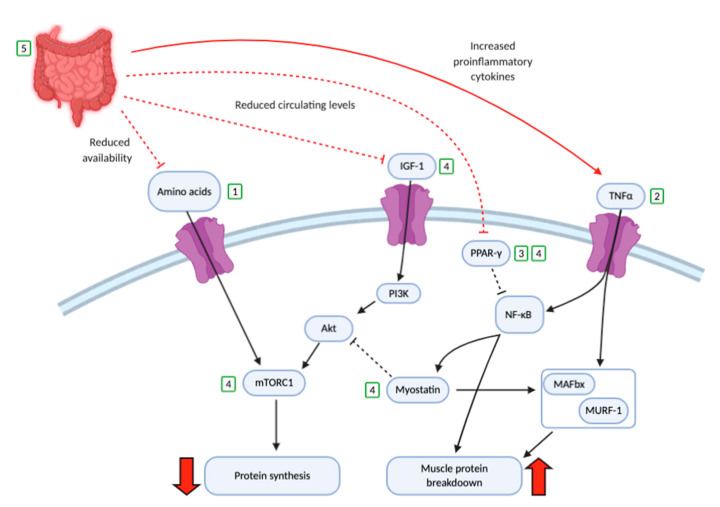
The possible impact of inflammatory bowel disease (IBD) on molecular pathways of muscle protein synthesis and muscle protein breakdown. Red arrows/lines demonstrate negative changes that occur in IBD. The green numbered squares demonstrate possible sites of intervention such that (1) protein supplementation, (2) anti-tumour necrosis factor alpha (TNFα) agents (3) omega 3 polyunsaturated fatty acids (*n*-3 PUFAs), (4) exercise programmes and (5) surgical intervention. Insulin growth factor-1 (IGF-1), peroxisome proliferator-activated receptor gamma (PPAR-γ), phosphoinositide 3-kinase (PI3K), nuclear factor kappa-light-chain-enhancer of activated B cells (NF-kB), protein kinase B (Akt), mammalian target of rapamycin complex 1 (mTORC1), muscle atrophy box (MAFbx) and muscle ring finger-1 (MuRF-1).

**Table 1 nutrients-13-00656-t001:** Overview of the tools used to assess sarcopenia in IBD.

Component	Muscle Mass	Muscle Function	Nutritional Assessment Tools
**Validated tools**	CT L3 CSA CT L3 SMI DXA ASMI	Nil	Nil
**Future research**	MRI Ultrasound	HGS Gait speed Timed up and go	MUST NRS 2002 NRI MIRT SaskIBD-NRT

Computerised tomography (CT), 3rd lumber vertebrae (L3), Cross sectional area (CSA), Dual-energy X-ray absorptiometry (DXA), Appendicular skeletal muscle index (ASMI), Magnetic resonance imaging (MRI), Hand grip strength (HGS), Malnutrition universal screening tool (MUST), Nutritional risk screening (NRS), Malnutrition inflammation risk tool (MIRT), Saskatchewan Inflammatory Bowel Disease Nutrition Risk Tool (SaskIBD-NRT).

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
