# Peer review of "Sarcopenia in Inflammatory Bowel Disease: A Narrative Overview"

_nutrients, 2021, doi:10.3390/nu13020656_

Round 1

Reviewer 1 Report

Dear Authors following are my suggestion to improve your manuscript.

You have chosen a very interesting topic but it is not possible for me to review the quality of your study, because the method part of your review is missing entirely. First, describe clearly which review type you have performed and use a fitting guideline to report your method part adequately. There are guidelines for classical literature reviews, like by Green BN, Johnson CD, Adams A. Writing narrative literature reviews for peer-reviewed journals: secrets of the trade. J Chiropr Med. 2006;5(3):101-117. doi:10.1016/S0899-3467(07)60142-6, or for other reviews types, like PRISMA guidelines: http://www.prisma-statement.org/ and their extended versions  for e.g. scoping reviews: http://www.prisma-statement.org/Extensions/ There are further guidelines in the international literature. It depends on your review type and what you prefer to use.

Reviewer 2 Report

Sarcopenia is an important medical and social problem. There are many causes of secondary sarcopenia, including chronic inflammatory bowel diseases. The knowledge on this subjects is constantly being expanded.

This work well complements the report by Ryan F et al. (2018) and An HJ et al.(2020).

In my opinion the work is well prepared in terms of content and editorial.     The authors presented the results of research by many researches included 132 literature items. The article contains  a large amount of  information  on sarcopenia in inflammatory bowel diseases, contained in sections : Assessment of Sarcopenia; Mechanisms driving Sarcopenia; The management of Sarcopenia.

Nevertheless, I only propose to suplement the Mechanisms driving Sarcopenia section with knowledge on role of intestinal bacteria( as minor revision).

It has been suggested that the gut microbiome could directly affect muscles by modulating amino acids bioavailability and the production of proinflammatory cytokines.

Reviewer 3 Report

This is an interesting topic of growing interest due to the high prevalence of Inflammatory Bowel Disease(IBS). Although in recent years several excellent reviews covering a very similar scope have been published  (e.g Ryan et al. Infl Bowel Diseas 2019) , this review is well written and organized and, especially, introduced explanations for contradicting results underlying the pathophysiology of sarcopenia in IBD.

Few comments :

  • In line 32  please define malnutrition according to the ESPEN definition (Cederhom et al. Clin Nutr 2017)
  • No data about the effects of enteral  nutrition /oral nutrition supplements on sarcopenia
  • Line 106, are you sure that aetiology of IBD is shown in Figure 2?
  • - In line 438 i am not quite sure about your statement that  all figures and tables should be cited"
  • What about the table in page 11? No title, no abbreviations, please explain
  • Please check references format according to journal style.

Round 2

Reviewer 1 Report

Dear Authors, I still miss the method part of your narrative review. It is not enough to include only a reference.
